# Preclinical Evaluation of the FGFR-Family Inhibitor Futibatinib for Pediatric Rhabdomyosarcoma

**DOI:** 10.3390/cancers15164034

**Published:** 2023-08-09

**Authors:** Jerry T. Wu, Adam Cheuk, Kristine Isanogle, Christina Robinson, Xiaohu Zhang, Michele Ceribelli, Erin Beck, Paul Shinn, Carleen Klumpp-Thomas, Kelli M. Wilson, Crystal McKnight, Zina Itkin, Hiroshi Sotome, Hiroshi Hirai, Elizabeth Calleja, Volker Wacheck, Brad Gouker, Cody J. Peer, Natalia Corvalan, David Milewski, Yong Y. Kim, William D. Figg, Elijah F. Edmondson, Craig J. Thomas, Simone Difilippantonio, Jun S. Wei, Javed Khan

**Affiliations:** 1Genetics Branch, Center for Cancer Research, National Cancer Institute, National Institutes of Health, Bethesda, MD 20892, USA; jerry.wu@nih.gov (J.T.W.); cheukt@gmail.com (A.C.); david.milewski@nih.gov (D.M.); yong.kim@nih.gov (Y.Y.K.); weij@mail.nih.gov (J.S.W.); 2Laboratory Animal Sciences Program, Frederick National Laboratory for Cancer Research, Frederick, MD 21702, USA; kristine.isanogle@nih.gov (K.I.); christina.robinson2@nih.gov (C.R.); difilips@mail.nih.gov (S.D.); 3National Center for Advancing Translational Sciences, Rockville, MD 20850, USA; xiaohu.zhang@nih.gov (X.Z.); michele.ceribelli@nih.gov (M.C.); erin.beck2@nih.gov (E.B.); shinnp@mail.nih.gov (P.S.); carleen.klumpp-thomas@nih.gov (C.K.-T.); kelli.wilson@nih.gov (K.M.W.); crystal.mcknight@nih.gov (C.M.); zina.itkin@nih.gov (Z.I.); craigt@mail.nih.gov (C.J.T.); 4Taiho Pharmaceutical Co., Ltd., Tsukuba 300-0034, Japan; h-sohtome@taiho.co.jp (H.S.); hiroshi-hirai@taiho.co.jp (H.H.); 5Taiho Oncology, Princeton, NJ 08540, USA; ecalleja@taihooncology.com (E.C.); vwacheck@taihooncology.com (V.W.); 6Clinical Pharmacology Program, National Cancer Institute, National Institutes of Health, Bethesda, MD 20892, USAnatalia.corvalan@nih.gov (N.C.);

**Keywords:** rhabdomyosarcoma, FGFR4, FGFR inhibitor, futibatinib, pediatric cancer

## Abstract

**Simple Summary:**

FGFR4 is a receptor tyrosine kinase overexpressed in rhabdomyosarcoma (RMS) and mutationally activated in ~10% of fusion-negative (FN) cases. The goal of this study was to evaluate the preclinical efficacy of futibatinib, a novel FGFR-family inhibitor, in treating RMS. We demonstrate that futibatinib is a potent inhibitor of FGFR4 and impedes growth of RMS cell lines expressing wild-type and mutant FGFR4 in vitro. We also show that futibatinib is synergistic with currently used chemotherapies against RMS in vitro. However, futibatinib is ineffective in RMS xenograft mouse models as a monotherapy and has a modest benefit when combined with currently used chemotherapeutics only in FGFR4-mutated RMS.

**Abstract:**

Rhabdomyosarcoma (RMS) is the most common pediatric soft tissue sarcoma. Despite decades of clinical trials, the overall survival rate for patients with relapsed and metastatic disease remains below 30%, underscoring the need for novel treatments. FGFR4, a receptor tyrosine kinase that is overexpressed in RMS and mutationally activated in 10% of cases, is a promising target for treatment. Here, we show that futibatinib, an irreversible pan-FGFR inhibitor, inhibits the growth of RMS cell lines in vitro by inhibiting phosphorylation of FGFR4 and its downstream targets. Moreover, we provide evidence that the combination of futibatinib with currently used chemotherapies such as irinotecan and vincristine has a synergistic effect against RMS in vitro. However, in RMS xenograft models, futibatinib monotherapy and combination treatment have limited efficacy in delaying tumor growth and prolonging survival. Moreover, limited efficacy is only observed in a PAX3-FOXO1 fusion-negative (FN) RMS cell line with mutationally activated FGFR4, whereas little or no efficacy is observed in PAX3-FOXO1 fusion-positive (FP) RMS cell lines with FGFR4 overexpression. Alternative treatment modalities such as combining futibatinib with other kinase inhibitors or targeting FGFR4 with CAR T cells or antibody-drug conjugate may be more effective than the approaches tested in this study.

## 1. Introduction

Rhabdomyosarcoma (RMS) is the most common pediatric sarcoma, with ~350 new cases in the United States each year [1]. Current treatment includes chemotherapy comprising of vincristine, actinomycin D, and cyclophosphamide (VAC) combined with local control by surgery, radiation, or both [2,3,4,5]. This cytotoxic regimen results in a relapse-free survival rate of over 70%, albeit with significant toxicity [6]. However, survival for patients with relapsed and metastatic disease is only 20–30% [6]. Moreover, despite decades of clinical trials, there have been no FDA-approved targeted treatments, meaning that the upfront treatment regimen for RMS has remained largely the same since the 1970s [5,7,8]. Novel treatments targeting the molecular pathways of RMS are needed to reduce treatment-related toxicity and improve outcomes for patients with high-risk diseases.

Fibroblast growth factor 4 (FGFR4) is a receptor tyrosine kinase implicated in the differentiation of myoblasts into skeletal muscle and is overexpressed in RMS [9]. In fusion-positive (FP) RMS, FGFR4 is a direct transcriptional target of the PAX3-FOXO1 fusion oncogene, which drives its elevated expression [10,11]. FGFR4 knockdown has been shown to reduce survival of FP RMS in vitro [12]. In fusion-negative (FN) RMS, FGFR4 is highly expressed in nearly all tumors and mutationally activated in approximately 10% of cases [8,13,14]. These mutations, which occur typically at amino acids 535 and 550 of the tyrosine kinase domain, cause FGFR4 to be constitutively active and drive tumor growth and metastasis [15]. Taken together, these known properties highlight the importance of FGFR4 in RMS and suggest that FGFR4 inhibition is a promising strategy for treating RMS.

Futibatinib is a novel irreversible FGFR-family selective inhibitor that covalently binds to the FGFR ATP-binding pocket in the cytoplasmic kinase domain, which inhibits phosphorylation and downstream signaling. It inhibits FGFR1-4 with IC_50_ values of between 1.4 and 3.7 nM in an enzymatic inhibition assay [16]. In preclinical studies, futibatinib demonstrated antitumoral activity against gastric, breast, lung, endometrial, bladder, and multiple myeloma cell lines with FGFR aberrations [16]. In a phase 1 dose expansion study, futibatinib treatment demonstrated an objective response rate (ORR) of 13.7% for 197 patients with different types of advanced solid tumors with FGFR aberrations and demonstrated a manageable safety profile [17]. Of note, futibatinib was recently granted FDA accelerated approval for the treatment of FGFR2 fusion/rearrangement-positive intrahepatic cholangiocarcinoma, with an ORR of 42% [18].

In a previous study, the multi-kinase inhibitor ponatinib was shown to inhibit mutant FGFR4-driven RMS tumor growth in mice [19]. However, despite FDA approval of ponatinib for Philadelphia-positive acute lymphoblastic leukemia and chronic myeloid leukemia, clinical development for RMS has not been pursued due to its multi-kinase activity and significant toxicities. In this study, we aimed to investigate the activity of futibatinib, which is a pure FGFR inhibitor and has a more manageable toxicity profile, in models of FP and FN FGFR4-mutant RMS [20]. Furthermore, we aimed to determine whether the addition of futibatinib to currently used chemotherapy regimens conveys a benefit in delaying tumor progression and improving survival in RMS xenograft models. 

## 2. Materials and Methods

### 2.1. Futbatinib Sensitivity for FGFR4 Mutants

Mutant FGFR4 expression vectors were constructed using site-directed mutagenesis. Mutant and wild-type constructs were transfected into HEK293T cells using Lipofectamine 3000 (Thermo Fisher Scientific, Waltham, MA, USA) according to the manufacturer’s protocol. Transfected cells were treated with futibatinib for 1 h, and cell lysates were analyzed for FGFR4 phosphorylation. For the detection of phosphorylated FGFR4 by ELISA, a DuoSet IC Kit (human phospho-FGFR4, R&D Systems Inc., Minneapolis, MN, USA) was used according to the manufacturer’s protocol.

### 2.2. Cell Culture and Reagents

Ba/F3 TEL-FGFR4, a murine hematopoietic cell line that expresses a constitutively active FGFR4 kinase domain [19], was cultured in RPMI-1640 medium supplemented with 10% fetal bovine serum (FBS), 2 mM L-glutamine, 1% penicillin/streptomycin, and 1.5 ug/mL puromycin. RMS559 is a FN RMS cell line with an FGFR4^V550L^ activating mutation; RH4 and SCMC are FP RMS cell lines with FGFR4 overexpression; and 7250 is a normal human fibroblast cell line. These cell lines were grown in DMEM medium supplemented with 10% FBS, 2 mM L-glutamine, and 1% penicillin/streptomycin. All cell lines were STR profiled and negative for mycoplasma. Futibatinib was obtained through a Cooperative Research and Development Agreement (CRADA) with Taiho Oncology (Princeton, NJ, USA). Irinotecan (752903, Pfizer, New York, NY, USA) and Vincristine (030906, Pfizer) were obtained through the NIH Division of Veterinary Resources Pharmacy.

### 2.3. Futibatinib Dose-Response and EC_50_ Calculations

EC_50_ of futibatinib for Ba/F3 TEL-FGFR4 cells was determined as previously described [19]. A total of 1 ng/mL of IL-3 was added to the RPMI medium to determine if the presence of IL-3 rescues Ba/F3 TEL-FGFR4 from cell death induced by FGFR4 inhibition. To determine the EC_50_ of futibatinib, RMS559, SCMC, and RH4 cells were seeded at a density of 4000 cells/well, and 7250 cells were seeded at a density of 2500 cells/well in 96-well plates, at a volume of 90 μL of culture medium. After overnight incubation, 10 μL of culture medium containing various concentrations of futibatinib was added. After 72 h, viability was quantified using CellTiter-Glo (Promega, Madison, WI, USA). Triplicated wells were used for each condition. Prism GraphPad 8 was used to calculate EC_50_.

### 2.4. Immunoblots

Cells were collected with a cell scraper, then pelleted by centrifugation, washed with ice-cold PBS, and lysed in RIPA buffer with 1% HALT Protease and Phosphatase inhibitors. Lysates were sonicated to shear genomic DNA and clarified by centrifugation at 18,000× *g* for 20 min. The protein concentration of the resulting supernatant was determined by BCA assay. 20 μg of protein was run on NuPage 4–12% BisTris gels (Invitrogen, Carlsbad, CA, USA) and transferred to PVDF membranes (Invitrogen). Membranes were blocked in 5% nonfat dried milk in TBST for 1 h at room temperature and incubated with primary antibody overnight at 4 °C. HRP-conjugated anti-rabbit secondary antibody was used to probe the primary antibody. Protein was visualized using Amersham ECL Western Blotting Detection Reagent (Amersham, UK). The following antibodies were used: pFGFR4 (ab192589, Abcam, Cambridge, UK), FGFR4 (8562, Cell Signaling Technology (CST), Danvers, MA, USA), pERK1/2 (4370, CST), ERK1/2 (4695, CST), and GAPDH (8884, CST). Relative band intensity was quantified in Image Lab software (Version 5.2.1, Bio-Rad, Philadelphia, PA, USA)

### 2.5. Drug Matrix Screen and Combination Experiment

For pairwise drug-combination assessments in matrix format, compounds were acoustically dispensed into a 1536-well white solid bottom tissue culture-treated plate (EWB041000A, Aurora Microplates, Whitefish, MT, USA) with an Echo 550 acoustic liquid handler (Labcyte, San Jose, CA, USA). A nine-point custom concentration range with a 1:2 serial dilution was used for each drug pair tested. Bortezomib (final concentration 20.3 µM) was used as a positive control for cell cytotoxicity. Cells were seeded into compound-containing plates at a density of 500 cells/well, in a final volume of 5 µL of growth media by using a Multidrop Combi dispenser (Thermo Fisher Scientific). Plates were covered by a stainless-steel gasketed lid to prevent evaporation and incubated for 48 h. At the 48-h time point, 3 µL of Cell Titer Glo (Promega) was added to each well using a BioRAPTR^®^ (Beckton Coulter, Brea, CA, USA) and plates were incubated at room temperature for 15 min with the stainless-steel lid in place. Luminescence readings were taken using a Viewlux reader (PerkinElmer, Waltham, MA, USA) with a 2 s exposure time per plate. The viability of compound-treated wells was normalized to DMSO, empty well controls were present on each plate, and combination-response plotting was automatically performed for each individual drug and drug combination. Each condition was replicated in 6 wells or 3 wells.

To confirm synergistic combinations found by the matrix screen, RMS559 and RH4 cells were seeded at a density of 10,000 cells/well in 96-well plates at a volume of 180 μL of culture medium. After 6 h of incubation, 20 μL of culture medium containing futibatinib combined with irinotecan or vincristine was added. Incucyte live cell imaging was used to track the confluence of the cells over 72 h. 

### 2.6. Caspase-Glo 3/7 Assay

RMS559 or RH4 cells were seeded at a density of 10,000 cells/well in 96-well plates at a volume of 90 μL of culture medium. After 6 h of incubation, 10 μL of culture medium containing futibatinib and chemotherapies was added. Cells were incubated for an additional 24 h before assay. 100 μL of Caspase-Glo reagent (Promega) was added per well and incubated for 30 min. Each condition was replicated in 6 wells or 3 wells. Luminescence readings were taken using a Spark microplate reader (Tecan, Männedorf Switzerland). 

### 2.7. Animal Studies

Animal studies were approved by the Animal Care and Use Committee (ACUC) of the Frederick National Laboratory for Cancer Research (FNLCR). FNLCR is accredited by AAALAC International and follows the Public Health Service Policy for the Care and Use of Laboratory Animals. Animal care was provided in accordance with the procedures outlined in the “Guide for Care and Use of Laboratory Animals (National Research Council; 1996; National Academy Press; Washington, DC, USA). 

Animal studies were conducted with 6- to 8-week-old NOD SCID Gamma (NSG) mice (CCR Animal Resource Program, Bethesda, MD, USA). Five million cancer cells were injected subcutaneously into the right flank of each mouse. Weight and tumors were measured twice a week. Tumors were measured using a caliper and the following formula was used to calculate tumor volume: *V* = *π*6(*L* ∗ *W* ∗ *H*). Futibatinib was suspended in water with 0.5% hydroxypropyl methylcellulose and given via oral gavage. Irinotecan and vincristine were diluted in 0.9% saline and given via tail vein injection. Detailed dosing schedules are found in the legends of figures. When tumors reached 100 to 150 mm^3^, the mice were randomized based on tumor size and body weights using Studylog Animal Study Workflow Software (version 3.1.399.8) into treatment arms (*n* = 10 per arm). Humane endpoints were defined as tumors exceeding 1500 mm^3^ or body weight falling below 80% of weight at the start of the study. For mice designated for immunohistochemistry (IHC) studies, tumors were allowed to grow to 300 mm^3^. Futibatinib was given at 15 mg/kg via oral gavage once daily Monday through Friday for one week before the tumors were harvested. 

### 2.8. Pharmacokinetic Studies

For the futibatinib pharmacokinetics study, blood samples were taken by cardiac puncture from three tumor-bearing mice at each time point of 0, 0.5, 1, 2, and 4 h after 25 mg/kg oral drug dosing. Plasma was collected after 1000× *g* spinning in heparinized collection tubes at 4 °C for 10 min and then snap freeze in liquid nitrogen. Determinization of drug concentration in serum was described elsewhere [21]. Briefly, plasma samples were deproteinized and then filtrated by a glass-filter. Futibatinib concentration was determined by liquid chromatography-tandem mass spectrometry. The preclinical pharmacokinetic (PK) evaluation of futibatinib was conducted using noncompartmental methods, employing Phoenix WinNonlin v8.3 (Certara Corp, Cary, NC, USA), which was validated according to FDA 21CFR Part 11 regulations. The observed values for maximum plasma concentration (C_max_) and time of maximum plasma concentration (T_max_) were recorded. The area under the concentration-time curve (AUC) from time zero to the last quantifiable sample (AUC_last_) was calculated using the linear-up/log-down and from its estimation, other PK parameters such as volume of distribution and drug clearance were derived. The evaluated PK parameters following the single 25 mg/kg dose included the volume of distribution during the terminal phase (V_z_), systemic clearance (CL), and effective half-life (T_1/2_). 

### 2.9. Immunohistochemistry

Tissues were fixed for 48–72 h in 10% neutral buffered formalin, trimmed for histology, and embedded in paraffin, and serial 5 µm sections were created for IHC and H&E staining. All sections were reviewed by a pathologist. FGFR4 (Cell Signaling #8562), KI67 (Cell Signaling #12202), and Cleaved Caspase 3 (Cell Signaling # 9661) were stained using a Bond RX (Leica Biosystems, Wetzlar, Germany) autostainer with citrate buffer as antigen retrieval and the Bond Polymer Refine kit used for antibody detection. FGFR4 was diluted to 1:100 for 30 min, KI67 to 1:200 for 30 min, and Caspase 3 1:1600 for 60 min. ERK ½ (Cell Signaling # 4376) and Total ERK (Cell Signaling 4695) were diluted 1:200 and 1:2000 respectively with overnight 4C primary antibody incubations for each. Staining ERK ½ and total ERK followed Cell Signaling Technologies manual benchtop staining protocol. Positive IHC controls included evaluation of cell pellets of known high and low FGFR4 expression, human colon carcinoma, small intestine, and thymus. Negative controls included replacing the primary antibody with a nonspecific antibody from the same species and of the same isotype. Whole Slide Imaging was performed with an Aperio ScanScope XT (Leica) at 200× in a single z-plane. Thresholds for positivity were determined using known positive controls. Annotations were made to quantify IHC within viable tumors, excluding regions of necrosis, adjacent normal tissues, and areas containing artifacts, including edge stain artifacts.

### 2.10. Statistical Analyses

The significance of differences in confluence, caspase 3/7 activity, and tumor volume was determined with a student’s *t*-test. The significance of differences in median survival was determined with the Mantel-Cox test. Synergy was calculated using the Bliss Independence Model [22].

## 3. Results

### 3.1. FGFR4^V550L^ Is the Most Common FGFR4 Driver Mutation and Sensitive to Futibatinib

Analysis of genomic DNA from tumors of 515 FN RMS patients revealed FGFR4 activating mutations in 50 (9.7%) patients (Table 1) [8]. Of the patients with FGFR4 activating mutations, FGFR4^V550L^ was the most common mutation (78%). FGFR4^N535K^ was the second most common mutation (12%), followed by FGFR4^V550E^ (8%) and FGFR4^N535D^ (2%). In HEK293 cells engineered to express one of the four FGFR4 activating mutations, futibatinib was found to inhibit phosphorylation of FGFR4 at an IC_50_ of 130 nM for FGFR4^V550L^, 134 nM for FGFR4^N535K^, 8428 nM for FGFR4^V550E^, and 203 nM for FGFR4^N535D^ (Figure A1, Appendix B). These data suggest that futibatinib would be an effective inhibitor for 92% of RMS tumors harboring FGFR4 activating mutation.

### 3.2. Futibatinib Inhibits FGFR4 and Is Cytotoxic to Ba/F3 Cells Expressing TEL-FGFR4 Selectively through FGFR4 Inhibition

Ba/F3 is a murine, interleukin (IL)-3 dependent cell line that can be rendered IL-3 independent with the introduction of a constitutively active kinase [23,24]. Previously, our lab engineered Ba/F3 cells to express TEL-FGFR4, a fusion of the kinase domain of FGFR4 and the extracellular domain of TEL, which induces dimerization of FGFR4 and its autophosphorylation [19]. Using the Ba/F3 TEL-FGFR4 cell line, we confirmed that futibatinib is a potent inhibitor of FGFR4 phosphorylation resulting in cell death (Figure A2). The addition of IL-3 rescued Ba/F3 TEL-FGFR4 cells in the presence of futibatinib, suggesting that futibatinib kills Ba/F3 TEL-FGFR4 cells selectively through FGFR4 inhibition.

### 3.3. FN RMS Cell Line with an FGFR4^V550L^ Activating Mutation Is More Sensitive to Futibatinib Treatment Than FP RMS Cell Lines

To determine the sensitivity of RMS cell lines to futibatinib treatment, we measured FGFR4 expression and phosphorylation using western blot, cell growth using Incucyte live cell imaging, and viability using CellTiter-Glo. RMS559, which has an FGFR4^V550L^ activating mutation [25], had the highest FGFR4 expression among the three RMS cell lines tested and was the only cell line with phosphorylated FGFR4 (Figure 1A). This indicates that FGFR4 is constitutively active in RMS559 but not RH4 or SCMC. Indeed, RMS559 was the most sensitive cell line to futibatinib treatment, with an EC_50_ of 0.48 µM (Figure 1B). RH4 and SCMC were substantially less sensitive to futibatinib treatment, with EC_50_s of 6.64 µM and 11.7 µM respectively (Figure 1B). The remarkable sensitivity of RMS559 to futibatinib was likely due to the high FGFR4 expression level and the presence of the FGFR4 activating mutation (Figure 1C).

To investigate the effect of futibatinib on the FGFR4 signaling pathway in these cells, we performed western blot analysis on RMS559 treated with increasing concentrations of futibatinib. In RMS559, futibatinib inhibited phosphorylation of FGFR4 and a downstream target ERK1/2 in a dose-dependent manner (Figure 1D). These results suggest that FGFR4 inhibitor monotherapy may be effective for RMS with FGFR4 activating mutations, but not for those without such mutations.

### 3.4. Futibatinib Monotherapy Is Ineffective in RMS Xenograft Models

Next, we tested Futibatinib in mouse xenograft models of RMS559, RH4, and SCMC. We expected that futibatinib would be particularly effective at delaying tumor growth and prolonging survival for the RMS559 xenograft model, but less effective for RH4 and SCMC xenograft models. We first performed a pharmacokinetic study in mice inoculated with SCMC and found that futibatinib has a half-life of approximately 4 h (Figure A3). We then performed an efficacy study testing varying dosages of futibatinib (5–25 mg/kg, daily (SID) in the three RMS xenograft models (Figure 2A). Surprisingly, we found that futibatinib had no significant effect in delaying tumor growth or prolonging survival (Figure 2B–D) in any of the xenograft models, regardless of dosage. IHC analysis for p-MAPK, a downstream target of FGFR4, in RMS559 tumors found that p-MAPK was suppressed in mice treated with futibatinib compared to vehicle (approaching significance, *p* = 0.072; Figure A6). This suggests that although futibatinib is pharmacodynamically active in suppressing FGFR4 signaling in the tumor, it is ineffective as a monotherapy for RMS, possibly due to adaptive resistance mechanisms to FGFR4 inhibition.

### 3.5. Combination of Futibatinib with Currently Used Chemotherapeutics Has a Synergistic Effect against RMS In Vitro

Due to the limited efficacy of futibatinib as a single agent against RMS xenograft models, we hypothesized that combining futibatinib with irinotecan or vincristine, chemotherapies currently used in the clinic for RMS [5,26], would be more effective in RMS models than any of the single drugs. To test this hypothesis, we first conducted a matrix viability experiment combining futibatinib with irinotecan or vincristine in RMS559 cells. Using a Bliss independence model, we found that there are multiple concentrations in which the futibatinib-irinotecan and futibatinib-vincristine combinations are synergistic (Figure 3A). To determine if a similar synergistic effect exists in RH4, we measured the confluence of RMS559 and RH4 treated with futibatinib-irinotecan and futibatinib-vincristine combinations. For RH4, a higher concentration of futibatinib was used due to RH4 being less sensitive to the drug. At 72 h post-treatment, we found that cells treated with the drug combinations had significantly lower confluence than those treated with the individual drugs (Figure 3B). To determine if the lower confluence resulted from apoptosis induced by the treatments, we measured cleaved caspase 3/7 activity at 24 h post-treatment. For both cell lines, futibatinib treatment resulted in a significant increase in caspase 3/7 activity compared to DMSO control (Figure 3C). Moreover, for both cell lines, futibatinib-vincristine treatment resulted in a significant increase in caspase 3/7 activity compared to either drug alone (Figure 3C). Futibatinib-irinotecan treatment did not result in a significant increase of caspase 3/7 activity compared to futibatinib alone for RMS559; however, this treatment did result in a significant increase of caspase 3/7 activity in RH4. These data suggest that the combination of futibatinib with irinotecan or vincristine is synergistic in delaying RMS progression through induction of apoptosis.

### 3.6. Futibatinib-Vincristine Combination Is Modestly Effective in the RMS559 Xenograft Model

Because futibatinib-irinotecan and futibatinib-vincristine combinations showed synergistic activity against RMS in vitro, we hypothesized that these combination treatments would be more effective at delaying tumor progression and prolonging survival compared to monotherapy in RMS xenograft models. First, a tolerability study was performed to determine the maximum tolerated dosages for the drug combinations, which were 15 mg/kg for futibatinib, 50 mg/kg for irinotecan, and 0.4 mg/kg for vincristine (Figure A4). The dosage of irinotecan was later adjusted to 25 mg/kg, with the details provided in the Figure 4 legend. Using these dosages, mice bearing RMS559, RH4, or SCMC tumors were treated with a three-week cycle of futibatinib-irinotecan, futibatinib-vincristine, individual drugs, or vehicle control (Figure 4A). In RMS559 xenograft models, futibatinib-vincristine had a small but significant effect in delaying tumor progression compared to vincristine monotherapy. After 18 days of treatment, the average tumor volume for the futibatinib-vincristine combination group was 769.0 mm^3^ compared to 1016 mm^3^ for the vincristine monotherapy group (0.76-fold, *p* = 0.028) (Figure 4B). The futibatinib-vincristine combination treatment also had a small but significant benefit in prolonging survival, with a median of 26 days, compared to vincristine monotherapy, with a median of 23 days (*p* = 0.047) (Figure 4C). At day 18 of therapy, the futibatinib-irinotecan group had an average tumor volume of 619.3 mm^3^ compared to 823.6 mm^3^ for irinotecan monotherapy (0.75-fold); however, this difference was not significant (*p* = 0.127) (Figure 4B). The futibatinib-irinotecan combination did not have a significant benefit in prolonging survival compared to irinotecan monotherapy (Figure 4C). In SCMC and RH4 xenograft models, neither futibatinib-irinotecan nor futibatinib-vincristine combinations had significant benefits in delaying tumor growth and prolonging survival compared to irinotecan and vincristine monotherapies (Figure 4D). These data suggest that the combination of futibatinib with currently used chemotherapies conveys a small benefit for RMS with an FGFR4 activating mutation, but little or no benefit for RMS without such mutation.

### 3.7. Triple Combination of Futibatinib, Irinotecan, and Vincristine Has Limited Benefit over Irinotecan-Vincristine Treatment

Since the futibatinib-irinotecan and futibatinib-vincristine combinations show a small benefit compared to monotherapy in RMS559 xenograft models, we hypothesized that a triple combination of futibatinib-irinotecan-vincristine would further delay tumor growth and prolong survival than dual combinations. Tolerability studies were performed to determine the maximum tolerated dosages for the three compounds, and dosages of 10 mg/kg for futibatinib, 6 mg/kg for irinotecan, and 0.4 mg/kg for vincristine were chosen for the study (Figure A5). Using these dosages, mice bearing RMS559 tumors were treated with a three-week cycle of futibatinib-irinotecan-vincristine, irinotecan-vincristine, futibatinib-irinotecan, futibatinib-vincristine, individual drugs, or a vehicle negative control (Figure 5A). However, the futibatinib-irinotecan-vincristine treatment showed a marked improvement in delaying tumor progression and prolonging survival compared to treatment of vehicle, monotherapies, or futibatinib-irinotecan/vincristine combinations. However, it did not have a significant advantage over the irinotecan-vincristine combination, indicating the benefit came largely from irinotecan and vincristine rather than futibatinib (Figure 5B,C). As was the case in the two-drug combination study, futibatinib-irinotecan and futibatinib-vincristine combinations had small benefits over monotherapy, but these benefits were not significant (Figure 5B). Of note, tumors grew substantially faster in this study than in the previous two animal studies. Whereas RMS559 tumors in the vehicle group reached an average of 1500 mm^3^ on day 18 of treatment for the two-drug combination study (Figure 4B), vehicle group tumors in this study reached 1500 mm^3^ on day 11 (Figure 5B) which resulted in shorter median survivals compared to the first two animal studies. Interestingly, in this experiment, futibatinib monotherapy had a significant benefit in delaying tumor growth compared to vehicle treatment, which was not observed in either of the previous two animal studies. At day 11 of treatment, mice in the futibatinib treatment group had an average tumor volume of 1072 mm^3^ compared to 1532 mm^3^ in the vehicle treatment group (0.70-fold, *p* = 0.035). The futibatinib group also had a longer median survival at 15 days compared to 11 days for a vehicle; however, this difference was not significant (*p* = 0.20).

## 4. Discussion

In this study, we investigated the efficacy of futibatinib, a pan-FGFR inhibitor, for treating RMS. We reported that futibatinib potently inhibits wild-type FGFR4 and 92% of FGFR4 activating mutations (N535D, N535K, V550L, and V550E) detected in FN RMS. We showed in vitro that futibatinib effectively inhibits phosphorylation of FGFR4 and decreases the viability of RMS559, a FN RMS cell line with an FGFR4^V550L^ activating mutation leading to FGFR4 phosphorylation. We also showed that combining futibatinib with irinotecan or vincristine is synergistic in decreasing cell viability and increasing apoptosis for RMS559 and RH4, an FP RMS cell line with a high FGFR4 expression. However, our data from RMS mouse xenografts indicated that futibatinib monotherapy is not likely an effective treatment for RMS. Although futibatinib monotherapy showed a significant reduction of tumor size in the three-drug combination animal study (Figure 5B), it is difficult to compare tumor growth and survival across studies because RMS559 tumors grew substantially faster in the three-drug combination animal study than in other two animal studies (futibatinib single agent or two-drug combination studies). As these studies were performed sequentially, the batch-to-batch differences among mice may result in different tumor growth rates. Importantly, each animal study was internally controlled, allowing for the comparison of treatment groups within each experiment. Our study also demonstrated that the combination of futibatinib with other standard-of-care chemotherapeutics is modestly beneficial only for RMS bearing FGFR4^V550L^ mutation but not beneficial for FP RMS. We also found that a triple combination of futibatinib, irinotecan, and vincristine is not significantly better than the combination of irinotecan and vincristine for FN FGFR4^V550L^ RMS. 

For FN RMS FGFR4^V550L^ cell line RMS559, the futibatinib-vincristine combination conveyed a small but significant advantage over single agents. FGFR4 inhibition is likely a viable approach for this subtype of RMS. Fiorito et al. found that treating RMS559 xenograft models with FGF401, a more selective inhibitor of FGFR4 than futibatinib was effective at delaying tumor progression [27,28]. There are several important differences between this study and that by Fiorito et al. [27]. The first is the time point at which treatment was initiated. In the study by Fiorito et al. [27], treatment began when tumors reached 60 mm^3^, whereas we began treatment when tumors reached 100–150 mm^3^, approximately double the size of those in Fiorito’s study. Beginning treatment at a smaller tumor size may have resulted in improved drug penetrance and better drug response. Another difference is that mice received the drug twice daily (except on weekends) to ensure 24-h inhibition of FGFR4 signaling, whereas in our study, mice were treated only once daily. Indeed, dosing at a higher frequency, or dosing over the weekends to increase exposure to the drug, may increase the inhibition of FGFR4 signaling and suppress tumor growth. A further difference is that Fiorito et al. [27], used nude mice while our study used NSG mice. It is possible that higher doses of futibatinib could have been given and tolerated in nude mice, and additionally, the intact innate immunity may have played a role in inhibiting tumor growth [29]. Aside from FGF401, several other FGFR4-specific inhibitors have been developed, which may in the future become effective treatments against FGFR4^mutant^ RMS [28,30].

Neither futibatinib monotherapy nor combination therapies were effective for FP RMS mouse xenografts, in keeping with data that FGFR4 overexpression is not the only driver for FP RMS. Indeed, PAX3-FOXO1 establishes super-enhancers driving the expression of not only FGFR4, but also other RTKs including IGF1R, ALK, and MET [10,31]. Overexpression of these RTKs may contribute to the resistance mechanism by allowing FP RMSs to “rewire” their signaling networks through activation of these kinases [32]. As this was a major limitation of the current study, additional studies are ongoing in our lab combining FGFR4 inhibition with blockade of other kinases that may be involved in bypass resistance mechanisms. Moreover, as FGFR4 phosphorylation is undetectable in FP RMS cell lines in vitro, high FGFR4 expression likely plays only a minor role in driving this RMS subtype. Therefore, for RMS without FGFR4 activating mutations, rather than through suppressing its kinase activity, targeting FGFR4 as a tumor-associated antigen (TAA) using immunotherapy may be more effective. Indeed, antibodies targeting FGFR4 have been developed by our lab and others, paving the way for CAR T or antibody-drug conjugates-based therapies [33,34,35].

## 5. Conclusions

In summary, futibatinib is a potent FGFR-family inhibitor that effectively inhibits the growth of RMS cell lines in vitro, particularly for FN RMS with FGFR4^V550L^, the most common FGFR4 activating mutation. With our experimental conditions and tumor models, we provide evidence that futibatinib conveys a modest benefit, when given alone or when combined with vincristine for RMS559 xenograft models. However, there was no anti-tumoral activity observed for futibatinib as a single agent or when combined with irinotecan or vincristine for FP RMS xenograft models using our experimental conditions, likely due to compensatory signaling of other kinases parallel to or downstream of FGFR4. Future experiments will focus on optimizing futibatinib dosing schedules and alternative combination therapies. 

## Figures and Tables

**Figure 1 cancers-15-04034-f001:**
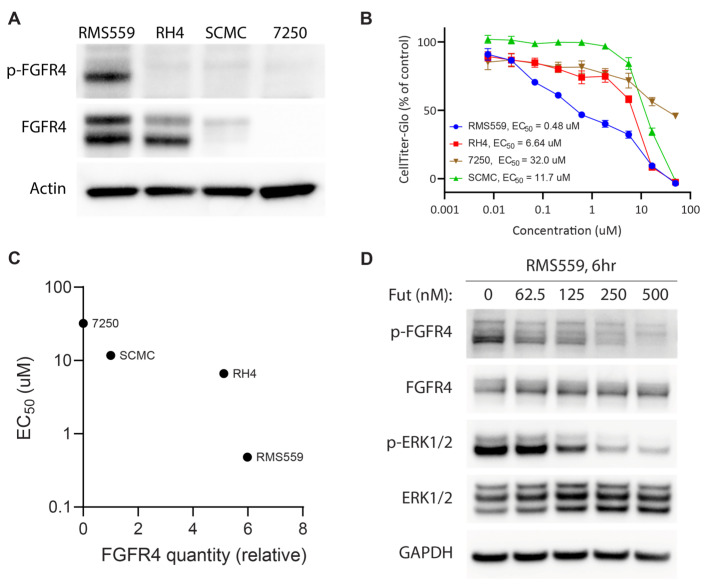
RMS559, an FN cell line with a FGFR4^V550L^ activating mutation, is more sensitive to futibatinib treatment than FP cell lines. (**A**) Western blot analysis of RMS559, RH4, SCMC, and 7250 (a negative control human fibroblast cell line). FGFR4 is phosphorylated only in RMS559, which has an FGFR4^V550L^ activating mutation. Phosphorylated FGFR4 was undetectable in FP RMS, nor 7250 cells. (**B**) Dose-response of cell lines using CellTiter-Glo at 72 h post-treatment shows marked sensitivity of RMS559 to futibatinib. Triplicated cells were used to generate the graph. (**C**) Relative FGFR4 quantity from (**A**) is plotted against futibatinib EC_50_. RMS559 is markedly more sensitive to futibatinib treatment compared to RH4. (**D**) Western blot of RMS559 treated with futibatinib for 6 h. Phosphorylation of FGFR4 and ERK1/2 is inhibited in a dose-dependent manner. See Appendix A for original Western Blots.

**Figure 2 cancers-15-04034-f002:**
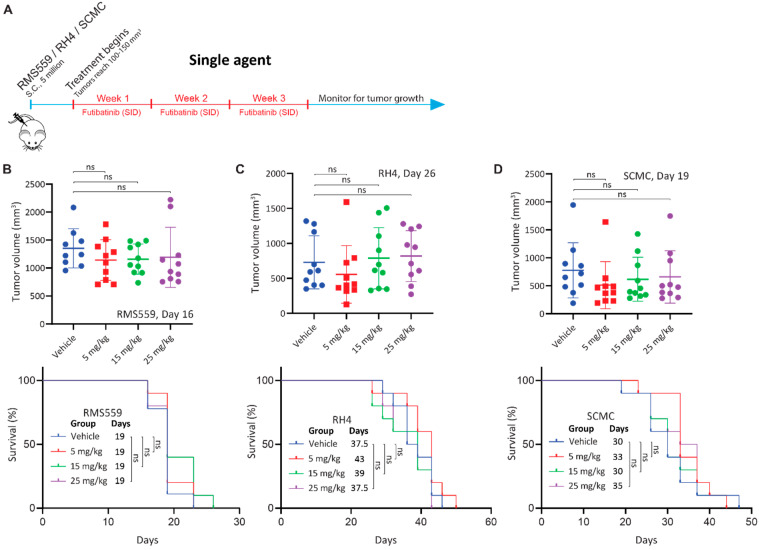
Futibatinib monotherapy is ineffective at inhibiting growth or prolonging survival in RMS xenograft models. (**A**) Schematic of the animal study testing futibatinib as a single agent. Mice (*n* = 10 per experimental arm) were inoculated with 5 million RMS559, SCMC, or RH4 cells via subcutaneous injection in the right flank. Treatment began when tumors reached 100–150 mm^3^ after randomization of mice into different treatment arms. Futibatinib or vehicle was given daily (SID) via oral gavage at 5, 15, or 25 mg/kg for three weeks. Mice were monitored for tumor progression until a humane endpoint was reached. (**B**–**D**) Tumor volume and survival of mice bearing RMS559, SCMC, and RH4 xenografts. The significance of tumor volume was determined by a student’s *t*-test. The significance of survival was determined by the Mantel-Cox test. ns, not significant.

**Figure 3 cancers-15-04034-f003:**
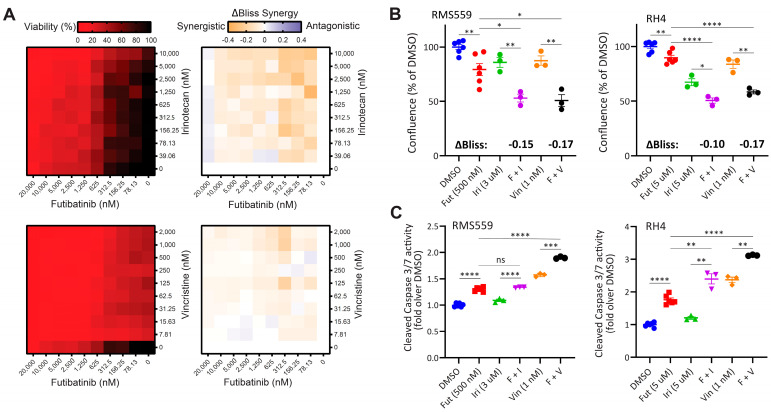
A combination of futibatinib with currently used chemotherapeutics has a synergistic effect against RMS in vitro. (**A**) Matrix plots (10 × 10) for the combinations of futibatinib (0–20 µM) with irinotecan (0–10 µM) or vincristine (0–2 µM) for viability (left, ATP assay) and ΔBliss (right) in RMS559 cells. ΔBliss matrices indicate multiple concentrations at which futibatinib-irinotecan and futibatinib-vincristine combinations are synergistic. (**B**) Incucyte confluence analysis for RMS559 and RH4 treated with futibatinib, irinotecan, vinciristine, futibatinib-vincristine, futibatinib-irinotecan, or DMSO at 72 h. * *p* < 0.05, ** *p* < 0.005 as determined by a student’s *t*-test. ΔBliss synergy scores for futibatinib-irinotecan and futibatinib-vincristine combinations are indicated in bold. Negative ΔBliss scores indicate synergy in reducing cell viability. (**C**) Caspase 3/7 activity assay for RMS559 and RH4 treated with futibatinib, irinotecan, vinciristine, futibatinib-vincristine, futibatinib-irinotecan, or DMSO at 24 h. * *p* < 0.05, ** *p* < 0.005, *** *p* < 0.0005, **** *p* < 0.00005 as determined by student’s *t*-test. ns, not significant.

**Figure 4 cancers-15-04034-f004:**
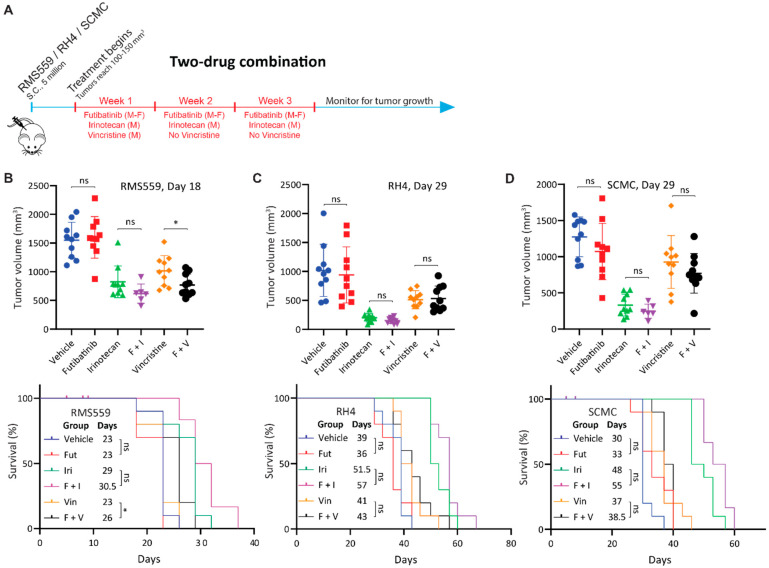
Futibatinib-vincristine combination treatment has a modest improvement over monotherapy in the RMS559 xenograft model. (**A**) Schematic of animal study testing futibatinib combined with irinotecan or vincristine. Mice (*n* = 10 per arm) were inoculated with 5 million RMS559, SCMC, or RH4 cells via subcutaneous injection in the right flank. Treatment began when tumors reached 100–150 mm^3^. Mice were dosed with futibatinib, irinotecan, vincristine, futibatinib-irinotecan, or futibatinib-vincristine. Futibatinib was given via oral gavage at 15 mg/kg, Monday to Friday, for three weeks. Vincristine was given once via tail vein injection at 0.4 mg/kg on the first Monday of the study. Irinotecan was given via tail vein injection initially at 50 mg/kg on Monday for three weeks. However, during the SCMC xenograft study, which was conducted before the RMS559 and RH4 xenograft studies, 50 mg/kg of irinotecan was found to be toxic. Thereafter, the dosage of irinotecan was adjusted to 25 mg/kg. Following cessation of dosing, mice were monitored for tumor progression until a humane endpoint was reached. (**B**) Tumor volume and survival of mice bearing subcutaneous RMS559 xenografts. On day 18, after post-initiation of treatment, tumors were significantly smaller in mice treated with futibatinib-vincristine compared to vincristine monotherapy (* *p* < 0.05, student’s *t*-test). Tumors were smaller in mice treated with futibatinib-irinotecan compared to irinotecan monotherapy, but this difference is not significant (*p* = 0.127, student’s *t*-test). Futibatinib-vincristine treatment confers a small but significant survival advantage compared to vincristine alone (* *p* < 0.05, Mantel-Cox test). In the futibatinib-irinotecan treatment group, four mice died due to toxicity near the beginning of the study; the events are censored. ns, not significant. (**C**,**D**) Tumor volume and survival of mice bearing SCMC and RH4 xenografts. During the SCMC xenograft study, four mice died in the futiatinib-irinotecan group due to toxicity; the events were censored. Thereafter, the dosage of irinotecan was decreased from 50 mg/kg to 25 mg/kg. For SCMC and RH4, futibatinib-irinotecan and futibatinib-vincristine do not confer a significant benefit in delaying tumor growth or prolonging survival compared to irinotecan and vincristine monotherapies. ns, not significant.

**Figure 5 cancers-15-04034-f005:**
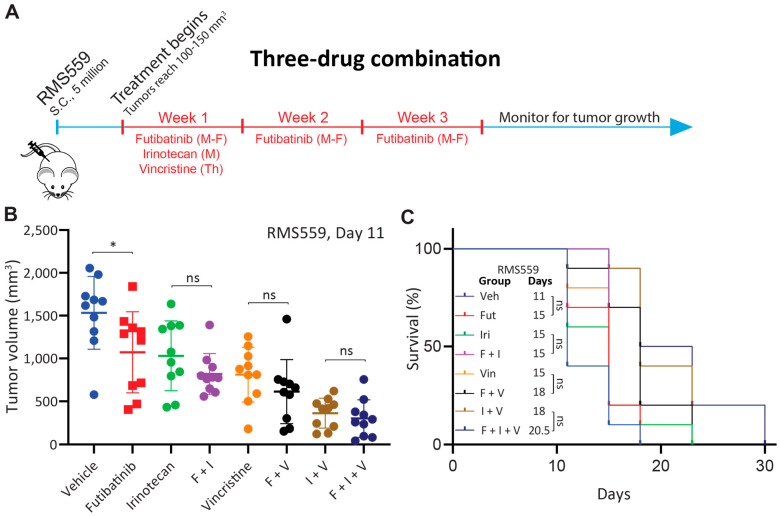
Futibatinib-irinotecan-vincristine combination treatment does not have a significant benefit over irinotecan-vincristine combination treatment in RMS559 xenograft model. (**A**) Schematic of animal study testing futibatinib-irinotecan-vincristine combination treatment. Mice were inoculated with 5 million RMS559 cells via subcutaneous injection in the right flank. Treatment began when tumors reached 100–150 mm^3^. Mice were dosed with futibatinib, irinotecan, vincristine, futibatinib-irinotecan, futibatinib-vincristine, irinotecan-vincristine, or futibatinib-irinotecan-vincristine. Futibatinib was given via oral gavage at 10 mg/kg, Monday to Friday, for three weeks. Irinotecan was given once via tail vein injection at 6 mg/kg on the first Monday of the study. Vincristine was given once via tail vein injection at 0.4 mg/kg on the first Thursday of the study. (**B**) Tumor volume on day 11 after initiation of treatment of mice bearing subcutaneous RMS559 xenografts. Although futibatinib-irinotecan-vincristine treatment substantially delayed tumor growth compared to vehicle, monotherapies, and futibatinib-vincristine and futibatinib-irinotecan combinations, it did not significantly delay tumor growth compared to the irinotecan-vincristine combination. Futibatinib monotherapy treatment significantly delayed tumor growth compared to vehicle, which was not observed in previous animal studies (* *p* < 0.05, student’s *t*-test.). ns, not significant. (**C**) Survival of mice bearing subcutaneous xenografts. Although futibatinib-irinotecan-vincristine treatment conferred a significant benefit to survival compared to vehicle, monotherapies, and futibatinib-vincristine and futibatinib-irinotecan combinations, it did not significantly improve survival compared to the irinotecan-vincristine combination. ns, not significant.

**Table 1 cancers-15-04034-t001:** Incidence of FGFR4 activating mutations in FN RMS and sensitivity of futibatinib against mutant and WT FGFR4.

FGFR4 Activating Mutation	Total Incidence, *n* (%)	Incidence among Activating Mutations	EC_50_ of Futibatinib (nM)
N535D	1/515, (0.2%)	2%	203
N535K	6/515, (1.2%)	12%	134
V550E	4/515, (0.8%)	8%	8428
V550L	39/515, (7.6%)	78%	130
Wild type			6

## Data Availability

Not applicable.

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
