# Peer review of "Preclinical Evaluation of the FGFR-Family Inhibitor Futibatinib for Pediatric Rhabdomyosarcoma"

_cancers, 2023, doi:10.3390/cancers15164034_

Round 1

Reviewer 1 Report

The Authors aimed to determine  whether the addition of futibatinib to currently used chemotherapy regimens conveys a  benefit in delaying tumor progression and improving survival in RMS xenograft models. 

The topic is interesting and the study well organized.

The paper is properly written and organized.

Introduction ok. I would detail further about the current evidence of treatment in RMS.

Did the Authors assessed different futinatinib doses as well as different intervals and/or associations?Please discuss.

I would use with caution the term "in vivo" as this is a xenograft study. Please check and correct as this might be misleading.

A part from these comments, very interesting and well conducted study.

Author Response

The Authors aimed to determine whether the addition of futibatinib to currently used chemotherapy regimens conveys a benefit in delaying tumor progression and improving survival in RMS xenograft models. 

The topic is interesting and the study well organized.

The paper is properly written and organized.

Introduction ok. I would detail further about the current evidence of treatment in RMS.

We thank the reviewer for their positive comments that the manuscript is interesting, organized, and well written. We have added references for the current combination chemotherapy regimen for RMS (Raney et al, J Clin Oncol 2011; 29:1312; Walterhouse et al, Cancer 2017; 123:2368; Arndt et al, J Clin Oncol 2009; 27:5182; Hawkins et al, J Clin Oncol 2018; 36:2770) and included a description in the manuscript.

Did the Authors assessed different futinatinib doses as well as different intervals and/or associations? Please discuss.

We thank the reviewer for raising this issue. We did assess different dosages of futibatinib at 5mg/kg, 10mg/kg, or 25mg/kg as part of our tolerability studies (figure S3). We found that 15mg/kg of futibatinib was the maximum tolerated dose when given in combination with chemotherapeutics. Moreover, once-daily dosing of futibatinib at 0.5-50mg/kg was previously found to be effective for multiple tumor xenograft models with aberrant FGFR signaling (Sootome et al, Cancer Res 2020,80:4986-97; Ito et al, ACS Med. Chem. Lett. 2023, 14:396-404). Therefore, inhibition of FGFR4 is expected to be achieved with the maximum tolerated dose used in our study.

Additionally, we have added to our manuscript pharmacodynamic markers of total and phosphorylated p44/p42 MAPK, which is a downstream target of FGFR4 signaling. We found a reduction of pMAPK:MAPK ratio in xenograft models treated with futibatinib 15mg/kg, which is suggestive of futibatinib’s activity. We have added this as figure S6 in the manuscript.

I would use with caution the term "in vivo" as this is a xenograft study. Please check and correct as this might be misleading.

We changed “in vivo” to mouse xenograft study.

Apart from these comments, very interesting and well conducted study.

We appreciate the reviewer’s comments and enthusiasm of this study, and hope our revision is now satisfactory to this reviewer.

Reviewer 2 Report

In the manuscript, the authors state that futibatinib is ineffective in vivo as a monotherapy and has a modest benefit when combined with currently used chemo therapeutics only in FGFR4-mutated RMS, however treatment modalities such as combining futibatinib with other kinase inhibitors or targeting FGFR4 with CAR T cells or antibody drug conjugate may be more effective than the approaches tested in this study..

The authors are conducting experiments using cultured Rhabdomyosarcoma (RMS) cell lines. Unfortunately, the results obtained from experiments using only culture cell lines are artificial findings and do not reflect the living body of cancer patients. At a minimum, the authors must conduct experiments using mouse models of developing RMS. In addition, the authors are required to perform validation using excised tissue obtained from a patient with RMS. In clinical practice, cancer genomic medicine using cancer gene panels, which is being conducted by medical teams around the world, has not shown high expression of PAX3-FOXO1 fusion in excised tissues of patients with RMS.

The authors should complete the manuscript by performing in vivo experiments in small animals and experimental systems using the resected tissues from lung cancer patients. Authors should submit the manuscripts to other journals.

In the manuscript, the authors state that futibatinib is ineffective in vivo as a monotherapy and has a modest benefit when combined with currently used chemo therapeutics only in FGFR4-mutated RMS, however treatment modalities such as combining futibatinib with other kinase inhibitors or targeting FGFR4 with CAR T cells or antibody drug conjugate may be more effective than the approaches tested in this study..

The authors are conducting experiments using cultured Rhabdomyosarcoma (RMS) cell lines. Unfortunately, the results obtained from experiments using only culture cell lines are artificial findings and do not reflect the living body of cancer patients. At a minimum, the authors must conduct experiments using mouse models of developing RMS. In addition, the authors are required to perform validation using excised tissue obtained from a patient with RMS. In clinical practice, cancer genomic medicine using cancer gene panels, which is being conducted by medical teams around the world, has not shown high expression of PAX3-FOXO1 fusion in excised tissues of patients with RMS.

The authors should complete the manuscript by performing in vivo experiments in small animals and experimental systems using the resected tissues from lung cancer patients. Authors should submit the manuscripts to other journals.

Author Response

In the manuscript, the authors state that futibatinib is ineffective in vivo as a monotherapy and has a modest benefit when combined with currently used chemo therapeutics only in FGFR4-mutated RMS, however treatment modalities such as combining futibatinib with other kinase inhibitors or targeting FGFR4 with CAR T cells or antibody drug conjugate may be more effective than the approaches tested in this study.

In clinical practice, generally new agents are used in combination with existing effective therapies that are currently in use, hence the choice of vincristine and irinotecan. However, we agree with the reviewer that combination with other kinase inhibitors may be more effective than chemotherapy, and this is the subject of ongoing studies and is beyond the scope of this current manuscript.

Moreover, we agree with the reviewer that other kinase inhibitors and CAR or antibody drug conjugates may be more effective. Indeed, these are the subject of ongoing studies and will be reported in future manuscripts. We have added this to the discussion.

The authors are conducting experiments using cultured Rhabdomyosarcoma (RMS) cell lines. Unfortunately, the results obtained from experiments using only culture cell lines are artificial findings and do not reflect the living body of cancer patients.

At a minimum, the authors must conduct experiments using mouse models of developing RMS. In addition, the authors are required to perform validation using excised tissue obtained from a patient with RMS. In clinical practice, cancer genomic medicine using cancer gene panels, which is being conducted by medical teams around the world, has not shown high expression of PAX3-FOXO1 fusion in excised tissues of patients with RMS.

The authors should complete the manuscript by performing in vivo experiments in small animals and experimental systems using the resected tissues from lung cancer patients. Authors should submit the manuscripts to other journals.

Cell line-derived xenograft (CDX) models are routinely used for drug efficacy research. We would like to point the reviewer to “the pediatric preclinical testing program: description of models and early testing results” (PMID: 17066459), which first describes the use of a list of pediatric cancer cell lines including RH4. Moreover, other studies have used RMS559 (PMID: 36097178) and SCMC (PMID: 28446439) CDXs to test the efficacy of drugs in animal models.

We agree that the PAX-FOXO1 fusion gene is generally not expressed at very high levels. However, the nature of transcription fusion gene-driven cancers is that the expression level, albeit modest, is sufficient to drive oncogenesis. This phenomenon is known as the “Goldilocks” principle in which the expression of the fusion transcription factor must be maintained at just-right levels because too much fusion protein is toxic, while too little fails to maintain the malignant properties. (PMIDs: 36505823, 34329586, and 36658220).

Given our modest results, we do not believe that testing combinations of futibatinib with chemotherapeutics in other RMS models will show greater efficacy.

Thus, we agree with the reviewer’s comments that there are no perfect models to test drug therapies in cancers. Furthermore, we agree that FGFR4 mutation-driven RMS may show benefits from more potent and specific inhibitors of the aberrant protein, and we have modified the manuscript accordingly.

Reviewer 3 Report

In this manuscript by Wu et al. the authors investigate the activity of futibatinib, a novel FGFR inhibitor, in rhabdomyosarcoma expressing FGFR4 wt (FP-RMS) or mutated (FN-RMS). The authors look at 4 different mutations at two sites, and find ¾ respond to inhibitor at low nanomolar concentrations. As monitored by phospho-FGFR4 ELISA.

In the cell line, Ba/F3, they confirms that activity is mediated by inhibition of FGFR4 phosphorylation, as cell death can be rescued by IL3.

Importantly, they observe that only in the FN-RMS cell line RMS559 is constitutive phosphorylated, suggesting that the inhibitor might be active only in this subset. In a model. In fact, RMS559 express the most FGFR4, are the only one phosphorylated / constitutively active – and exibit the lowest EC50 for futibatinib 0.48uM.

Next, they test futibatinib monotherapy in RMS559 FGFR4mut (V550L) (FN), RH4 FGFR4wt (FP) and SCMC FGFR4wt (FP). Three different doses (5, 15, 25 mg/kg) are tested, but no activity is observed. Through in vitro screening, they identify combinations with vincristine or irinotecan that act synergistically with futibatinib, even in the FP RH4 cell line. These results are confirmed by increased caspase 3/7 activity. In vivo, only the combination futibatinib/vincristine in RMS559 FGFR4mut has an increased efficacy compared to the single treatments, although quite modest. All other combinations do not show synergistic effect in vivo.

Finally, the authors try a triple therapy with RMS559. Unfortunately, the addition of futibatinib to irinotecan/vincristine does not show a significant advantage.

In the discussion, the authors address all the limitations of this important study, and conclude that the size of the tumors chosen to start the studies (100-150 mm^3) might have reduced tumor penetration of futibatinib compared to a previous study by Fiorito et al where 60 mm^3 was chosen. They speculate that in the future, other inhibitors might be more effective for FN-RMS with mutant FGFR4, while for FP-RMS, they don’t expect FGFR4 to deliver any therapeutic benefit since it might rely on other RTK to survive. Here, other RTK inhibitors combinations need to be explored.

This manuscript is very well written, and the result solid and well presented. The study is carefully designed, and the conclusions are on point. The topic addresses it of great importance for FGFR-dependent entities, which are not restricted to pediatric rhabdomyosarcoma.

As a generally remark, I would like to have the author’s comment, on how is it possible to confirm that the inhibitor is active in vivo, that the concentration reached was similar to what used in culture, and how would it be possible to monitor, e.g. on tumor sections, if FGFR4 phosphorylation is reduced (downstream target analysis?).

As a minor point, the author should describe how futibatinib concentration was determined in plasma. Was it done/tried in tumor tissue as well?

Author Response

In this manuscript by Wu et al. the authors investigate the activity of futibatinib, a novel FGFR inhibitor, in rhabdomyosarcoma expressing FGFR4 wt (FP-RMS) or mutated (FN-RMS). The authors look at 4 different mutations at two sites, and find ¾ respond to inhibitor at low nanomolar concentrations. As monitored by phospho-FGFR4 ELISA.

In the cell line, Ba/F3, they confirms that activity is mediated by inhibition of FGFR4 phosphorylation, as cell death can be rescued by IL3.

Importantly, they observe that only in the FN-RMS cell line RMS559 is constitutive phosphorylated, suggesting that the inhibitor might be active only in this subset. In a model. In fact, RMS559 express the most FGFR4, are the only one phosphorylated / constitutively active – and exibit the lowest EC50 for futibatinib 0.48uM.

Next, they test futibatinib monotherapy in RMS559 FGFR4mut (V550L) (FN), RH4 FGFR4wt (FP) and SCMC FGFR4wt (FP). Three different doses (5, 15, 25 mg/kg) are tested, but no activity is observed. Through in vitro screening, they identify combinations with vincristine or irinotecan that act synergistically with futibatinib, even in the FP RH4 cell line. These results are confirmed by increased caspase 3/7 activity. In vivo, only the combination futibatinib/vincristine in RMS559 FGFR4mut has an increased efficacy compared to the single treatments, although quite modest. All other combinations do not show synergistic effect in vivo.

Finally, the authors try a triple therapy with RMS559. Unfortunately, the addition of futibatinib to irinotecan/vincristine does not show a significant advantage.

In the discussion, the authors address all the limitations of this important study, and conclude that the size of the tumors chosen to start the studies (100-150 mm^3) might have reduced tumor penetration of futibatinib compared to a previous study by Fiorito et al where 60 mm^3 was chosen. They speculate that in the future, other inhibitors might be more effective for FN-RMS with mutant FGFR4, while for FP-RMS, they don’t expect FGFR4 to deliver any therapeutic benefit since it might rely on other RTK to survive. Here, other RTK inhibitors combinations need to be explored.

This manuscript is very well written, and the result solid and well presented. The study is carefully designed, and the conclusions are on point. The topic addresses it of great importance for FGFR-dependent entities, which are not restricted to pediatric rhabdomyosarcoma.

We thank the reviewer for recognizing the importance of this study even with its negative results.

As a generally remark, I would like to have the author’s comment, on how is it possible to confirm that the inhibitor is active in vivo, that the concentration reached was similar to what used in culture, and how would it be possible to monitor, e.g. on tumor sections, if FGFR4 phosphorylation is reduced (downstream target analysis?).

We thank the reviewer for bringing up this important point. We performed immunohistochemistry on tumors harvested from RMS559 xenograft models treated with futibatinib, staining for H&E, FGFR4, phosphorylated and total p44/p42 MAPK. We attempted to test phosphorylated FGFR4 antibodies, but none of the commercial antibodies showed specific staining. MAPK is a downstream target of FGFR4 and can be used as a proxy for FGFR4 activity. The ratio of phosphorylated MAPK to total MAPK appeared to be suppressed in xenograft models treated with futibatinib, asthis difference was approaching to significance (p=0.072, ) due to the limited number of mice used to perform IHC (n=2). The IHC data suggest the inhibitor was active in vivo, and we have included them in Figure S6 of the manuscript.

In addition, previously published efficacy of in vivo mice pharmacodynamic studies support that sufficient exposure level of futibatinib for inhibition of FGFRs was achieved in xenograft tumors at doses of 5 to 25 mg/Kg(Sootome, H. et. Al., Cancer Res., 80, 4986-4997, 2020 and Ito, S. et al, ACS Med. Chem. Lett. 14, 396-404, 2023, Supplementary information 2.6).

As a minor point, the author should describe how futibatinib concentration was determined in plasma. Was it done/tried in tumor tissue as well?

We thank the reviewer for recognizing this important detail. Detailed materials and methods of mouse pharmacokinetic study was described elsewhere (Ito, S. et al, ACS Med. Chem. Lett. 14, 396-404, 2023, Supplementary information 2.6). Briefly, the isolated plasma samples were deproteinized, then filtrated by glass-filter. Futibatinib concentration was determined by LC-MS. We have added this to our revision.

As stated above, although we did not determine the drug concentrations in tumor tissues, efficacy results of in vivo mice pharmacodynamic studies support that sufficient exposure level of futibatinib for inhibition of FGFRs was achieved in xenograft tumors at doses of 5 to 25 mg/Kg(Sootome, H. et. Al., Cancer Res., 80, 4986-4997, 2020 and Ito, S. et al, ACS Med. Chem. Lett. 14, 396-404, 2023, Supplementary information 2.6).

Reviewer 4 Report

In the present work by Wu et al. the mechanisms of FGFR-family inhibitor futibatinib, are presented. The authors reported on the treatment of rhabdomyosarcoma (RMS). Their work is interesting, as RMS is a devastating disease and as rare tumor, much research is needed in order to understand the mechanics behind RMS biology. The present work is in the right direction as it addresses a very interesting tumor.

The present work is interesting and it has merit for publication, after some minor corrections.

The authors reported that FGFR inhibitors are not very effective as a mono-therapy, but are more effective when combined with other chemotherapeutics. This is known, in general, for inhibitors (for example the authors could mention the case of gefitinib, which is also a FGF inhibitor and cells develop resistance, due to mutational occurrences). The authors should suggest possible mechanisms for this phenomenon. How is this to explain with respect to RMS biology and response to chemotherapy?

Interestingly, the authors mention that bortezomib was used as positive control. Since bortezomib is a proteasome inhibitor, how do the authors account for the difference in the cellular and RMS response? How do they account for the differences between bortezomib and futibatinib?

I suggest to mention the limitations of the study. What can be done better, or what are the next steps?

Finally, the authors should highlight their results and mention how their findings could prove useful for the treatment of RMS. How is it possible to use combinatorial therapy in order to treat RMS?

Author Response

In the present work by Wu et al. the mechanisms of FGFR-family inhibitor futibatinib, are presented. The authors reported on the treatment of rhabdomyosarcoma (RMS). Their work is interesting, as RMS is a devastating disease and as rare tumor, much research is needed in order to understand the mechanics behind RMS biology. The present work is in the right direction as it addresses a very interesting tumor.

The present work is interesting and it has merit for publication, after some minor corrections.

We thank the reviewer for recognizing the importance of this study.

The authors reported that FGFR inhibitors are not very effective as a mono-therapy, but are more effective when combined with other chemotherapeutics. This is known, in general, for inhibitors (for example the authors could mention the case of gefitinib, which is also a FGF inhibitor and cells develop resistance, due to mutational occurrences). The authors should suggest possible mechanisms for this phenomenon. How is this to explain with respect to RMS biology and response to chemotherapy?

We appreciate the insight of the reviewer concerning the ineffectiveness of monotherapy of FGFR4 inhibition. It is possible that other FGFR4 mutations could occur the prolonged presence of futibatinib. Additionally, other resistance mechanisms such as compensatory activation of other kinases, due to feedback loops, including activation of IGF1R, and SRC pathways that have been reported to be active in RMS. However, this is beyond the scope of this study and is the subject of ongoing studies. In the discussion section of the manuscript, we have added comments emphasizing the possible mechanisms of resistance.

Interestingly, the authors mention that bortezomib was used as positive control. Since bortezomib is a proteasome inhibitor, how do the authors account for the difference in the cellular and RMS response? How do they account for the differences between bortezomib and futibatinib?

We would like to clarify that bortezomib was used as a positive technical control for cytotoxicity in the drug combination screen, as a vast majority of cell lines are sensitive to this drug. It was not used as a control for futibatinib due to their very different mechanisms of action.

I suggest to mention the limitations of the study. What can be done better, or what are the next steps?

Finally, the authors should highlight their results and mention how their findings could prove useful for the treatment of RMS. How is it possible to use combinatorial therapy in order to treat RMS?

We agree with the reviewer. There are several limitations of this study.

  1. Inhibitors that specifically target FGFR4 mutations may be more effective in RMS and are the subject of studies by our lab and others.
  2. Compensatory upregulation in other TK pathways, which can be targeted by combinations with other TKIs.
  3. Drug schedule can be optimized, such as continuous dosing, including weekends or twice daily, due to the relatively short half-life of futibatinib, although this could introduce additional toxicity.

We thank the reviewer for the insights and suggestions. We have revised the discussion accordingly and believe the comments improved the manuscript. 

Reviewer 5 Report

1. I did not find statistical information in the Materials and Methods section: in how many repetitions were studies conducted on cell lines? In Figure 1B, C, the "whiskers" are not visible; we would like to evaluate the reproducibility of the results and the error of the experiment. The same applies to experiments on mice: how many animals participated, provide data.

2. Since futibatinib has shown efficacy in cell lines but not in tumor xenografts, is the delivery of the drug to the tumor a problem?

Author Response

  1. I did not find statistical information in the Materials and Methods section: in how many repetitions were studies conducted on cell lines? In Figure 1B, C, the "whiskers" are not visible; we would like to evaluate the reproducibility of the results and the error of the experiment. The same applies to experiments on mice: how many animals participated, provide data.

We apologize for this omission, and should have included this information in the methods. Three replicated wells were used for generation of IC50s in Figure 1B. For those data points where whiskers are not visible, it is because the error bars are smaller than the size of the symbols, a known characteristic of Graftpad Prism (https://www.graphpad.com/support/faqid/2041/). We added this information in the Method. We thank the reviewer for his/her careful review.

  1. Since futibatinib has shown efficacy in cell lines but not in tumor xenografts, is the delivery of the drug to the tumor a problem?

We agree with the reviewer that delivery of the drug could be an issue. However, previous studies have shown that once daily dosing of 5-20mg/kg was effective for multiple human tumor xenograft models driven by FGFRs (Sootome et al, Cancer Res 2020,80:4986-97; Ito et al, ACS Med. Chem. Lett. 2023, 14:396-404). We agree that drug schedule can be optimized, such as continuous dosing, including weekends or twice daily, due to the relatively short half-life of futibatinib, although this could introduce additional toxicity.

We have also added figures in Supplemental Figure S6 to show that FGFR4 downstream markers of total and phosphorylated p44/p42 MAPK, demonstrated a reduction of pMAPK:MAPK ratio using futibatinib at 15mg/kg. The difference was close to significance (p=0.072), due to the limited number of mice used to perform IHC (n=2). All this evidence supports the inhibitory activity of futibatinib in mouse xenograft tumors using our dosing schedule, despite its ineffectiveness. We thank the reviewer for giving us this opportunity to clarify and have included it in the revision.